# Synthesis and Characterization of a Fe_3_O_4_@PNIPAM-Chitosan Nanocomposite and Its Potential Application in Vincristine Delivery

**DOI:** 10.3390/polym13111704

**Published:** 2021-05-23

**Authors:** Cynthia N. Hernández-Téllez, Ana G. Luque-Alcaraz, Maribel Plascencia-Jatomea, Hiram J. Higuera-Valenzuela, Mabeth Burgos-Hernández, Nadia García-Flores, Mario E. Álvarez-Ramos, Jorge L. Iriqui-Razcon, Pedro A. Hernández-Abril

**Affiliations:** 1Ingeniería Biomédica, Universidad Estatal de Sonora, Hermosillo 83100, Mexico; cynthia.hernandez@ues.mx (C.N.H.-T.); ana.luque@ues.mx (A.G.L.-A.); hiram.higuera@ues.mx (H.J.H.-V.); jorge.iriqui@ues.mx (J.L.I.-R.); 2Departamento de Investigación y Posgrado en Alimentos, Universidad de Sonora, Hermosillo 83000, Mexico; maribel.plascencia@unison.mx; 3Licenciatura en Ecología, Universidad Estatal de Sonora, Hermosillo 83100, Mexico; mabeth.burgos@ues.mx; 4Departamento de Física, Universidad de Sonora, Hermosillo 83000, Mexico; nadian.garciafl@gmail.com (N.G.-F.); enrique.alvarez@fisica.uson.mx (M.E.Á.-R.)

**Keywords:** drug delivery, nanocomposite, chitosan, PNIPAM, vincristine, magnetite

## Abstract

In this research, we conducted a systematic evaluation of the synthesis parameters of a multi-responsive core-shell nanocomposite (Fe_3_O_4_ nanoparticles coated by poly(N-isopropylacrylamide) (PNIPAM) in the presence of chitosan (CS) (Fe_3_O_4_@PNIPAM-CS). Scanning electron microscopy (SEM) was used to follow the size and morphology of the nanocomposite. The functionalization and the coating of Fe_3_O_4_ nanoparticles (Nps) were evaluated by the ζ-potential evolution and Fourier Transform infrared spectroscopy (FTIR). The nanocomposite exhibited a collapsed structure when the temperature was driven above the lower critical solution temperature (LCST), determined by dynamic light scattering (DLS). The LCST was successfully shifted from 33 to 39 °C, which opens the possibility of using it in physiological systems. A magnetometry test was performed to confirm the superparamagnetic behavior at room temperature. The obtained systems allow the possibility to control specific properties, such as particle size and morphology. Finally, we performed vincristine sulfate loading and release tests. Mathematical analysis reveals a two-stage structural-relaxation release model beyond the LCST. In contrast, a temperature of 25 °C promotes the diffusional release model. As a result, a more in-depth comprehension of the release kinetics was achieved. The synthesis and study of a magnetic core-shell nanoplatform offer a smart material as an alternative targeted release therapy due to its thermomagnetic properties.

## 1. Introduction

Nanometric scale research has opened a promising perspective on understanding the behavior of matter at this level. It has been observed that, at regular or macro size, the properties of a material with the same chemical composition are different than when presented as nanostructures [1]. Its main outstanding property is the increase in surface contact because hundreds of atoms are located on the surface of the nanostructures, available to react per square centimeter. Recently, these nanostructures and their properties are beginning to be used regularly, as they can be applied in a wide range of disciplines. In modern medicine, the application of nanostructures in different forms has been proposed: nanoparticles as biomarkers for imaging [2], magnetic hyperthermia [3], arrays as drug delivery systems [4], charged magnetic particles proposed as targeted therapy, and drug release as is the case in this study. The development of modern technology for its practical application and performance requires intelligent materials to innovate and lead different challenges such as biophysics [5], aerospace [6], biotechnology [7], biomedicine [8], among other current challenges [9,10], versatile materials that represent a challenge for nanoscience and nanotechnology. Among the smart materials, CS can be highlighted, widely known, and applied in biomedicine due to its biocompatibility, biodegradability, and adhesiveness, among other favorable properties. CS is an N-deacetylated derivative of chitin, which is commercially present in different degrees of deacetylation and different molecular weights. Due to its basic character and a 6.2–7.0 pK, it is insoluble in many aqueous solutions, making it challenging to apply in neutral or alkaline environments. That is why CS can undergo different modifications to increase its physicochemical properties such as solubility, porosity, and architectural forms, among others [11].

On the other hand, PNIPAM is an intelligent thermosensitive material because it can change its structural conformation under temperature variations of a few degrees Celsius (32 °C approximately), known as its LCST, originating new properties of the material undergoing these temperature changes. This characteristic can be exploited and applied as micelles or vesicles for drug delivery [4]. Increasing the LCST of the material close to the organism’s body temperature (37 °C approximately) allows a better-controlled release once internalized as a drug release platform. Increasing the LCST of PNIPAM can be achieved by adding a polymer such as CS to the nanoplatform, increasing the material’s biodegradability since PNIPAM shows high chemical stability, thus achieving fractional degradation [12].

One of the most significant challenges of drug or radioisotope delivery treatments in the organism is to avoid damage to healthy cells and reduce toxicity [13]. The inclusion of magnetite (Fe_3_O_4_) to a nanoplatform gives it the property of bio-directing [14] and locating the particle towards a specific tissue or cells [15]; in addition, it is well tolerated by the organism and susceptible to biopolymer coatings to be biocompatible [16]. All of the above allows biomedicine to use Fe_3_O_4_ by hyperthermia, a complementary therapeutic procedure based on the increase in temperature in a tumor mass using this type of nanostructures; Jasso-Terán et al. reported an increase in temperature of magnetic nanoparticles of 41.2 °C, simulated under physiological conditions in vitro [17].

One of the most important aspects of a drug delivery platform is biocompatibility [18]. Multiple studies report high toxicity of magnetite [19,20]; however, as they are coated with biocompatible polymers such as PNIPAM and CS, this parameter is increased [21,22]. Kean and Thanou (2010) report that when supplying CS and its derivatives intravenously, the distribution route is initially in the liver and lung during the first 60 min [23], then in these organs the enzymes such as lysozyme and chitin start the degradation of this polymer [24]. Briceño et al., (2017) report a study of magnetite nanoparticle degradation in a biomimetic medium; this study shows that nanoparticles that were not covered with OA show rapid degradation in the first 24 h, while those that were coated with OA show slow degradation of up to 480 h [25]. For all the above, these materials show desirable characteristics of biocompatibility and biodegradability in a platform for drug delivery.

Additionally, by internalizing and precisely localizing a nanoplatform in the organism, a compound can be released into a target tissue to perform a specific action [26,27,28]. In this study, vincristine coupled to the nanocomposite is proposed as cancer therapy and targeted release. Vincristine is one of the fundamental cancer treatments, due to its well-defined mechanism of action, in addition to its proven anticancer activity. Its effectiveness against childhood leukemia has been proven in potentially fatal childhood hemangiomas [29,30]. Silverman & Deitcher (2013) reported that injecting vincristine in the form of a liposome-based on sphingomyelin nanoparticles and cholesterol to the organism exceeds the dosage pharmacokinetics of standard vincristine, delivering an amount of active drug to the target tissues and showing superior antitumor activity [31]. The synthesis and study of a thermo-magnetic core-shell nanoplatform represent a smart thermosensitive material as an alternative targeted release therapy in the organism due to its thermomagnetic properties.

## 2. Materials and Methods

### 2.1. Materials

The materials used were: N,N’-methylenebisacrylamide (BIS) 154.17 g/mol (99.5%), N-isopropylacrylamide (NIPAM) 113.16 g/mol (98%), ammonium persulfate (APS) 228.20 g/mol (98%), sodium dodecyl sulfate (SDS) (≥99%), iron(II) chloride (FeCl_2_) (≥98%), iron(III) chloride (FeCl_3_) (97%), oleic acid (OA) (≥99%), and sodium hydroxide (NaOH) (≥98%). Chemical reagents were obtained from Sigma-Aldrich (St. Louis, MO, USA). Deionized water (18.25 MΩ/cm) was used in all experiments. Nitrogen gas was obtained from a local supplier.

### 2.2. Synthesis of NpFe_3_O_4_

The NpFe_3_O_4_ was synthesized by the iron salts coprecipitation method by adding NaOH under oxygen-free environments [21]. A solution of FeCl_3_ and FeCl_2_ (2:1 molar ratio) was placed into 20 mL deionized water. The solution was mechanically stirred at 700 rpm while nitrogen gas flowed into the flask for 30 min, resulting in the oxygen’s displacement. The solution temperature rose to 70 °C; after 15 min, the color turned yellow to wine-red. The next step was adding 10 mL of NaOH (30%), turning the color from wine-red to black, indicating the formation of magnetite Nps. Afterward, OA was added to the reaction solution in a 1:1.5 molar ratio (FeCl_3_: OA), correspondingly. The OA was exposed to Nps for 15 min to achieve their stabilization. Lastly, the synthesized Nps were separated from the solution using a neodymium magnet. The solution was separated, and a cleaning process was carried out, first with pure ethanol, then with a 50% ethanol–water mixture, and ultimately with pure water resulting in a shiny black material.

### 2.3. NIPAM Polymerization on NpFe_3_O_4_ in Presence CS (Fe_3_O_4_@PNIPAM-CS)

In the polymerization, the method of precipitation by free radicals was followed [32]. In the first step, 150 mmol of NIPAM was dissolved in 10 mL of ultrapure water. Next, 10 mmol of BIS and 20 mmol of SDS were added. Np_0_ is a sample without CS and NpFe_3_O_4_. Np_1_, Np_2_, and Np_3_ are the samples with NpFe_3_O_4_ and molar ratios 1:0, 1000:1, 1000:2, and 1000:3 (NIPAM-CS), respectively. After this, nitrogen gas flowed in the solution flask for 25 min, providing an oxygen-free surrounding. These solutions were warmed to 70 °C. Finally, 1 mL of 5 mM APS was carefully added. The nitrogen stream and magnetic agitation conditions were maintained for 30 min at 70 °C. An opalescent solution without polymer additions was observed. The resulting samples were centrifuged fractionally, first at 3000 RPM for 30 min (separating the uncoated NpFe_3_O_4_), then the resulting supernatant at 5000 RPM for 25 min. The resulting pellet was resuspended in ultra-pure water for posterior analyses.

### 2.4. Characterization

The morphologies of Np_0_, Np_1_, Np_2_, and Np_3_ were evaluated with the Scanning Electron Microscope (SEM) (JEOL JSM-7800F, Pleasanton, CA, USA) at 3.0 kV. The samples were diluted 1 to 15 before deposition on the film. The size distribution of NpFe_3_O_4_, Np_0_, Np_1_, Np_2_, and Np_3_ samples was determined by DLS using a Zetasizer-Nano ZS (Malvern Instruments, Malvern, UK), with a laser of wavelength = 633 nm (He–Ne, 4.0 mW). The ζ-potential of all samples was determined using a sizer-nano ZS (Malvern Instruments, Malvern, UK). The samples’ magnetization properties were carried out at 300 K using a Physical Property Measurement System vibrating Sample Magnetometer (PPMS-VSM, Quantum Design, San Diego, CA, USA); the samples were previously lyophilized. The vincristine sulfate load and release in Fe_3_O_4_@PNIPAM-CS were measured by indirect quantification using a Perkin Elmer UV-Vis spectrometer (lambda 850, Perkin Elmer Inc., Waltham, MA, USA); the absorbance was fixed at 310 nm. FTIR spectra were collected from a Spectrum (Perkin Elmer, Inc., Waltham, MA, USA) spectrometer equipped with a single diamond attenuated total reflectance (ATR) with a range of 8300–350 cm^−1^ variable IR beam with diameter from 2 to 11 mm. The IR spectra were collected 16 times (spectral resolution 4 cm^−1^).

### 2.5. Vincristine Sulfate Loading

The load vincristine sulfate was determined by subtraction of the initial vincristine solution. The curve of vincristine standards (0–100 µg/mL) was constructed to estimate the amount of free drug in the wash solution. Encapsulation efficiency was determined using the function of vincristine sulfate encapsulated and the initial amount of Fe_3_O_4_@PNIPAM-CS.

### 2.6. Vincristine Sulfate Release Study Mathematical Release Model

The vincristine sulfate release pathway was elucidated by the dialysis tube method. The release mechanism was studied under different temperature conditions. In the first step, Fe_3_O_4_@PNIPAM-CS-Vincristin pellet was resuspended in 10 mL of PBS Buffer at a temperature of 37 °C. Sampling was carried out for a prolonged period to observe the details of the release mechanism. All the samples taken were replaced with freshly prepared PBS buffer solution.

Two mathematical models were fitted to the resulting data from the three temperatures. Model 1 is the power-law equation used for releases with a Fickian system [33] (Equation (1)).
(1)$$\frac{M_{t}}{M_{\infty }}=k_{1}t^{n}$$

*M∞* represents the water intake at the time of equilibrium. The constant *k*_1_ is the front factor of the swelling ratio. Finally, *n* is the swelling exponent related to the water sorption mechanism.

Model 2 describes the behavior revealed by the data at 37 and 41 °C; at these temperatures, two release stages are evident; therefore, the BiDoseResp equation was used (Equation (2)).
(2)$$\frac{M_{t}}{M_{\infty }}=A_{1}+\left(A_{1}-A_{2}\right)*\left\{\left(\frac{p}{1+10^{\left(logx01-x\right)\left(h_{1}\right)}}\right)+\left(\frac{p-1}{1+10^{\left(logx02-x\right)\left(h_{2}\right)}}\right)\right\}$$

This equation was used to model the resistivity as a function of an alloy’s temperature [34] and used to describe the fluorescence enhancement behavior with the increase of pH [35]. Using this model in the release of vincristine is necessary to correlate the parameters of the equation with the physical phenomena involved in the release kinetics. The parameters h_1_ and h_2_ are related to the speed of the release burst in the first and second stage, respectively, and they are directly proportional to the slope of the linear part of the stage. Parameters *A*_1_ and *A*_2_ are related to the duration of the first and second stage, respectively. *logx*01 and *logx*02 represent the proportion released during the plateau between the two stages; the addition of their absolute values is directly proportional to the fraction released during the stage transition.

## 3. Results

### 3.1. Swelling Kinetics and LCST

In this research, the influence of CS concentration in the LCST was tracked and evaluated by the temperature profile using DLS observations. Pure PNIPAM Nps (Np_0_) shows a typical collapse induced by hydrophobic interactions in aqueous media at a temperature of 33 °C. The size of Np_0_ was about 260 nm for the swollen state (21 °C) and 103 nm in a collapsed state (43 °C). The Np_1_ sample contains the lowest amount of CS. A reduction of the LCST at 31 °C was observed, attributed to the OA on the Fe_3_O_4_ Nps. An illustrative schematic of the nanocomposite performance and the kinetics of swelling and collapse is shown in Figure 1.

The relatively low molar ratio of CS-PNIPAM allows the influence of OA to prevail. This reduction effect was reported in multiple works [8,36], and we have reported this effect due to the OA in a PNIPAM matrix [21]. In the Np_2_ and Np_3_ samples, we observed that the effect of CS predominates because of its greater concentration; the physical integration of the hydrophilic chains of CS leads to the alteration of the hydrophilic/hydrophobic equilibrium. The results indicate a displacement of the LCST to higher values for the samples Np_2_ and Np_3_ (33 and 39 °C respectively) (Figure 2). The rise of the LCST offers the possibility of focusing its application on releasing drugs into the human body whose internal temperature is around 37 °C. The maximum size in the swollen state (21 °C) of the particles was affected by the presence of CS. The size of the particles Np_1_, Np_2_, and Np_3_ at that temperature are 297, 292, and 280 nm, respectively (Figure 2). To explain this behavior, we propose a model in which the spherical particles reached a protonable group saturation stage (amine groups). In this sense, excess amine groups can lead to repulsion between the same groups and generate packaged polymeric structures with more significant spatial restriction. In this way, it is possible to establish the conditions to control the particles’ size and LCST as a function of the amount of CS present in the reaction environment. All the above allows us to address design issues concerning Nps with thermal sensitivity and establish the conditions for the internalization of nanoparticles in normal and cancer cell lines.

### 3.2. Infrared Spectroscopy Analysis

In the FTIR spectrum of the NpFe_3_O_4_ sample, a signal is observed around 1562 cm^−1^ which is related to the C=O stretching of a carboxylic acid. In addition, two bands belonging to the C=H_2_ group are observed at 2921 and 2853 cm^−1^ associated with its asymmetric and symmetric stretching, respectively (Figure 3b). The above peaks are characteristic of the OA with which NpFe_3_O_4_ is coated. The bands attributed to OA in the NpFe_3_O_4_ sample do not belong to magnetite because this technique is not sensitive to this compound above 600 cm^−1^. The above has been reported previously; for example, Chiung-Hua et al., (2020) presented IR spectra of Nps prepared with doxorubicin/gelatin/Fe_3_O_4_-alginate that showed some characteristic peaks of all components except magnetite for a range of 1000 to 4000 cm^−1^ [37]. A band is observed around 562 cm^−1^, attributed to the Fe-O vibration of NpFe_3_O_4_ (Figure 3b). The presence of bands related to the OA functional groups is the first indication of their correct incorporation on the NpFe_3_O_4_ surface. The FTIR analysis of the Np_0_ sample presents a band in the vicinity of 1630 cm^−1^; such a signal is related to the carbonyl stretching to the amide group of PNIPAM. Around 3285 cm^−1^, an absorption related to N-H stretching is present. Finally, a double band at 1366 and 1386 cm^−1^ due to the isopropyl group of PNIPAM is present (Figure 3a).

The signal related to Fe-O stretching is observed with lower intensity in samples Np_1_, Np_2,_ and Np_3_, which have the same amount of NpFe_3_O_4_ and demonstrates its correct incorporation into the platform (Figure 3). In samples Np_1_, Np_2_ and Np_3,_ a small shoulder is observed around 1063 cm^−1^, attributed to the C-O stretching of the CS pyranose ring. This shoulder increases in sample Np_3_, which contains the most CS. The increase in the signal is directly proportional to the addition of CS, indicating that it is correctly incorporated into the system and that the cleaning processes do not eliminate it. In samples Np_1_, Np_2_ and Np_3,_ a small shoulder is observed at 1065 cm^−1^, attributed to C-O stretching of the CS pyranose ring. This shoulder is higher in sample Np_3_, which contains more CS (Figure 3e). The signal increases as more CS is added in its correct incorporation into the system and as the clean-up processes did not remove it.

### 3.3. Magnetic Properties

Figure 4 presents the hysteresis curves measured at 300 K of the Fe_3_O_4_Nps and Np_3_ samples. The magnetic saturation values are obtained around 34 and 9 emu/g for NpFe_3_O_4_ and Np_3_ samples, respectively. The magnetic properties are strongly correlated with the particle size [38]. Commonly, the superparamagnetic phenomena are exhibited for particles smaller than 10 nm [39]. Accordingly, considering the particle size achieved, superparamagnetic phenomena should be expected. The phenomenon is evident in the NpFe_3_O_4_ sample, given that there is no coercivity, and regardless of the increment in size in the Np_3_ sample, the same behavior is observed. The obtained values are promising for potential use in biomedical applications; for example, Timur Atabaev et al., (2013) report 34.97 emu/g and 15.12 emu/g for bare and core-shell structured samples, respectively [40].

All mentioned above allows us to conclude that the superparamagnetic properties were not reduced by the agglomeration of small particles in the core-shell nucleus and enables the synthesized platform to potential biomedical applications in which this property is appreciated.

### 3.4. ζ-Potential

In our case, the magnetite Nps were coated with OA to improve their stability and biocompatibility. The ζ-potential of Fe_3_O_4_ Nps after the functionalization process was around 7.1 ± 0.3, 18.3 ± 0.4, and 24.8 ± 0.3 mV. The electrostatic interactions between the NIPAM monomer and the OA layer resulted in a complex formation. Polymerization of PNIPAM led to an essential change in particle size and ζ-potential. In this way, we confirmed the surface functionalization of NpFe_3_O_4_, and also the increase in the concentration of CS made it possible to obtain nanoparticles with a positive and higher ζ-potential (Figure 5), which benefits its colloidal stability of the nanoparticles and influences the effectiveness of its interaction with negatively charged cell membranes.

The results concorded with Shagholani et al., (2015), who indicated that Fe_3_O_4_ nanoparticles with a ζ-potential of +36.4 mV had increased positive values of the ζ-potential of CS- Fe_3_O_4_ value (+48.1 mV) caused by the CS addition, confirming that functionalization of amino groups on the Fe_3_O_4_ surface by CS molecules [41]. The results obtained in the FTIR analysis showed the presence of Fe_3_O_4_, CS, and PNIPAM in samples Np_1_, Np_2_, and Np_3_. NpFe_3_O_4_ has a negative character; when NIPAM polymerizes in the presence of NpFe_3_O_4_, a ζ-potential close to the isoelectric point is obtained. The above indicates that the PNIPAM character predominates on the surface of the particle. The addition of CS in the reaction results in a positive ζ-potential, and their increase directly proportional to the amount of CS in the reaction indicates that it is interpenetrated to PNIPAM on the surface of the Fe_3_O_4_ particle, forming a core-shell structure. The results indicate electrostatic solid repulsive forces between nanoparticles; this provides good colloidal stability of the suspension since a ζ-potential of this was desired to obtain a physically stable suspension. This positive charge facilitates binding nanoparticles with siRNA and DNA plasmids to form complex magnetic nanoparticles for targeting genes delivered into cells [42,43].

### 3.5. Morphology Analysis

The shape and size of the nanoparticles Np_0_, Np_1_, Np_2_, and Np_3_ were analyzed by SEM and are shown in Figure 6. It can be observed that the sizes of the particles are 208, 304, 272, and 210 nm for Np_0_, Np_1_, Np_2_, and Np_3_, respectively, all presenting a spherical morphology. According to the micrographs, the nanoparticles exhibit a similar shape and structure characteristic of the pure PNIPAM (Np_0_) nanoparticles in all treatments. Likewise, we can highlight from Figure 6 that in the images of Np_1_, Np_2_, and Np_3_ the particles present a defined core of magnetite and a shell of PNIPAM. The difference in particle sizes between the different treatments can be attributed to the increase in CS concentration by increasing the amount of CS chains. The phenomenon of steric hindrance occurs, which limits the polymerization of PNIPAM, resulting in particles with smaller sizes; this analysis coincides with that reported by Kang and Kim (2010), who explained that a lower concentration of CS derives a more expanded structure than with pure PNIPAM [44]. The addition of CS shows a trend in reducing the size of the particles, presenting an inversely proportional behavior, since the particle with the highest amount of CS is the one with the minor diameter (Np_3_ < Np_2_ < Np_1_) compared to that of pure PNIPAM. These observations coincide with what was previously reported by Yi Gong (2012), showing that incorporating CS polymer chains into the PNIPAM particle results in smaller diameter nanoparticles [45]. SEM micrographs allow visualizing in a general way a core-shell morphology; it is observed the lack of core in the Np_0_ sample, while a higher electron density is observed in the center in the samples containing magnetite. The combined FTIR, ζ-potential, and SEM results allow us to conclude that samples Np_1_, Np_2_, and Np_3_ exhibit a core-shell structure. It should be noted that the Np_3_ treatment (particles of smaller diameter) was selected to perform vincristine release analysis due to its characteristics to act as a release vehicle.

### 3.6. Drug Loading (DL) and Encapsulation Efficiency (%EE)

In this study, the Np_3_ sample was selected due to physicochemical properties to determine the Average Encapsulation Efficiency (%EE) and the the Drug Load (DL). The determination of %EE and DL was performed for five vincristine sulfate concentrations (10, 20, 30, 40 and 50 µg/mL). The concentration of polymers was constant (Figure 7). It is evident that when the vincristine sulfate concentration rises, the LD goes up, and the %EE goes down. At a 30 mg/mL concentration, a %EE of 89.2 ± 0.005 and a DL of 2.77 ± 0.05 are observed. Although no data on EE and DL of vincristine encapsulated in PNIPAM Nps exist in the literature, the EE values obtained are high-ranking compared to those reported for other systems [46,47]. Likewise, DL values are comparable with other systems based on PNIPAM [48]. The values obtained suggest the high efficiency of the encapsulation and the suitability of the nanoplatform for use in drug delivery applications [49]. This sample was selected for controlled release testing.

### 3.7. Effect of Temperature on Drug Delivery

Model 1 fits the data well at 25, 37, and 41 °C, and the value of R^2^ decreases as a function of temperature (Table 1). At 37 and 41 °C, the release kinetics exhibits two well-defined stages (Figure 8). Model 2 fits well at 37 and 41 °C. We relate the first release stage to the diffusion of hydrophilic groups out of the particle, a classical Fickian-type diffusion. At temperatures above the LCST (37 and 41 °C), the concept of polymeric relaxation becomes relevant; the initial burst (first stage) in the release is so fast that it does not allow the inner polymeric layers to break the hydrogen bridges and expel the liquid inside them, steric-trapped inside the Nps. As the polymeric chains allow the hydrophilic molecules to migrate to the particle’s surface, the second release stage occurs.

Experimental data at 37 °C shows R^2^ > 0.996 (Table 2) and R^2^ < 0.978 (Table 1) for model 2 and model 1, respectively. At 25 °C, it was not possible to adjust the data to model 2. Otherwise, the release fitted to model 1 R^2^ > 0.975 in the swollen state (25 °C). We relate this limitation because, at 25 °C, the colloidal system is below the LCST temperature (39 °C for this sample Np_3_). Therefore, the relaxation stage does not occur. Inversely, when increasing the experimental temperature and approaching the LCST, an adjustment for a two-stage model occurs.

The constant *k*_1_ is the front factor of the swelling ratio and, *n* is the swelling exponent related to the water sorption mechanism.

The terms *h*_1_ and *h*_2_ are related to the speed of the release burst in the first and second stage, respectively. *A*_1_ and *A*_2_ are related to the duration of the first and second stage, respectively. The terms *logx*01 and *logx*02 represent the proportion released during the plateau between the two stages.

For experiments conducted at 37 and 41 °C, data fitted for model 2 R^2^ > 0.996 and their parameters are presented in Table 2. The second release stage was promoted by structural relaxation of the polymer. Thereby, when the temperature approaches the LCST of the PNIPAM particle, the transitions from a swollen to a collapsed state result in the polymer chains’ relaxation. The release by the relaxation mechanism is carried out near the LCST transition temperature (32 °C) (Figure 8). We associate this change with developing the structural transition from a hydrophobic state to a collapsed arrangement. Increasing the temperature makes it evident that the colloidal system’s temperature exceeded the LCST point (Figure 8). Therefore, diffusional phenomena (burst type release) are promoted by energy changes in the system. When the release system was at 41 °C, the fraction released for the first stage (diffusional) was around 0.8. All the above demonstrates that the burst type release is promoted by temperature increment. The data at 37 and 41 °C show that the temperature near the LCST triggers the drug’s extended-release (Figure 8).

The study of temperature through the mathematical model is beneficial in applications for biomedicine like personalized chemical therapy. We can conclude that temperature modifications help control the drug release mechanism and its concentration in the release. The adequate regulation in the drug dosage and release mechanism are beneficial in the advances of biomedicine. Because it is possible to design personalized drug release therapies and long-acting drugs, the effect of this potentiation can be achieved by taking advantage of the virtues of nanotechnology by confining the drug in the vicinity of the nano-excipient.

## 4. Conclusions

In the present research, the CS concentration on the morphology, ζ-potential and the hydrodynamic diameter of Fe_3_O_4_@PNIPAM-CS systems were studied. When the polymerization’s CS concentration was increased, the particle size decreased from 304 to 210 nm, and the LCST of the polymeric matrix was shifted from 33 to 39 °C successfully. Excess amino groups’ contribution leads to inter-group repulsion and results in packaged polymeric structures caused by the spatial restriction. The electrostatic interactions between the NIPAM monomer and the OA layer yielded a complex formation. With the polymerization of NIPAM in the magnetic particles’ vicinity, a significant change in particle size and ζ-potential was obtained; this confirms its correct functionalization, resulting in a core-shell structure. The Fe_3_O_4_@PNIPAM-CS systems preserve their sensitivity to magnetic and temperature stimuli. With a spherical morphology and a size of 210 nm in the dehydrated state, the Np_3_ sample demonstrates a desired magnetic and thermal response. The change in its hydrodynamic radius indicates its strong ability to transport and release hydrophilic molecules.

The sample Np_3_ has a well-defined polymeric shell. A stimuli-responsive Np_3_ sample loaded with the antineoplastic vincristine sulfate was evaluated as a drug delivery system using the power-law model and a two-stage model. At 25 °C, the drug was released by unrestricted diffusion. On the other hand, at 41 °C, a structural relaxation mechanism occurred, affected by the structural change upon exceeding LCST. According to the above, a helpful nanoplatform has been designed for versatile chemical therapy that provides high customization according to the requirements.

## Figures and Tables

**Figure 1 polymers-13-01704-f001:**
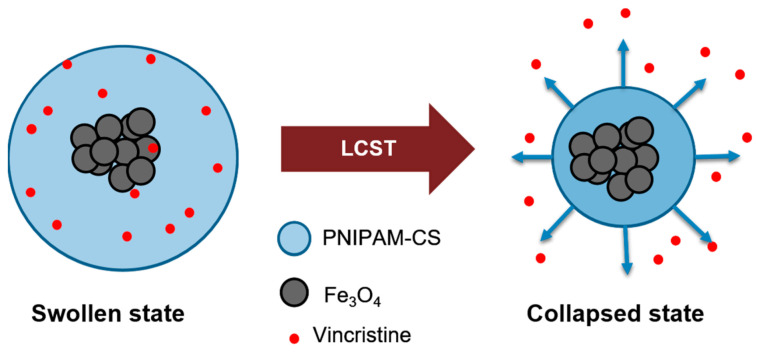
Illustrative schematic of the nanocomposite (Fe_3_O_4_@PNIPAM-CS-Vincristine) performance and the kinetics of swelling and collapse.

**Figure 2 polymers-13-01704-f002:**
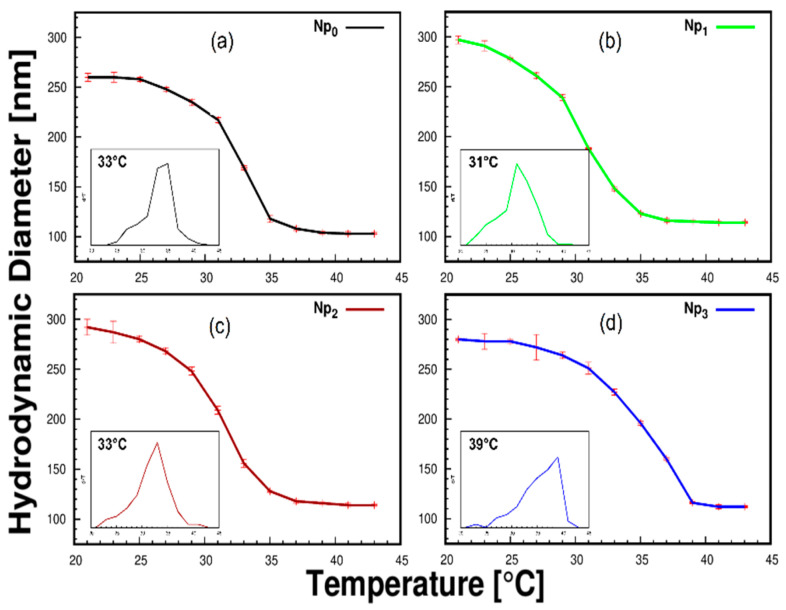
Effect of temperature and nanoparticle sizes of samples (**a**) Np_0_; (**b**) Np_1_; (**c**) Np_2_, and (**d**) Np_3_ in aqueous media, examined by dynamic light scattering measurements. The temperature values refer to the phase transition. The inset graphics represent the LCST behavior between nanoparticle size and temperature.

**Figure 3 polymers-13-01704-f003:**
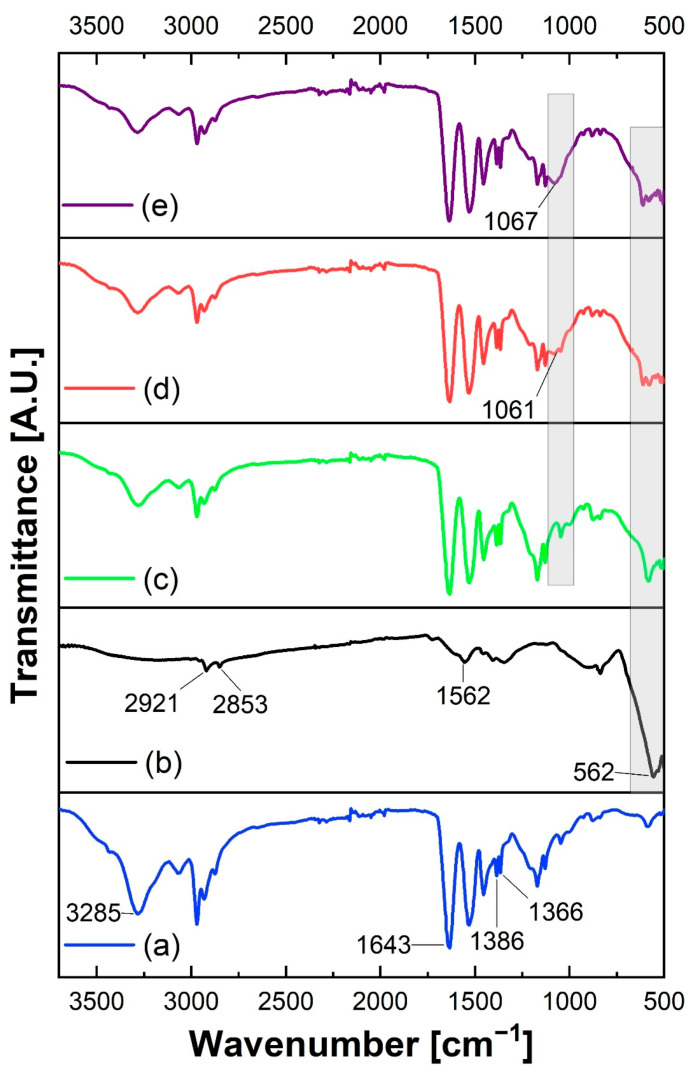
FTIR spectra of vacuum-dried powder of (**a**) Fe_3_O_4_Nps, (**b**) Np_0_, (**c**) Np_1_, (**d**) Np_2_, and (**e**) Np_3_. They were collected over the wavenumber range of 4000 to 500 cm^−1^.

**Figure 4 polymers-13-01704-f004:**
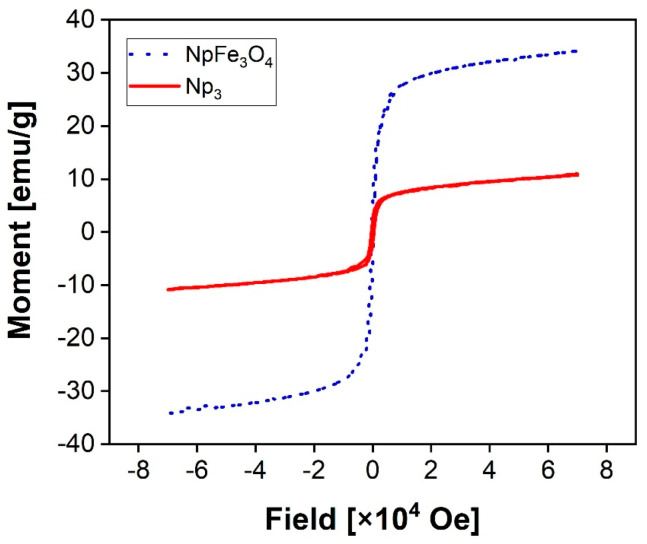
The magnetic curve of NpFe_3_O_4_ and Np_3_ at 300 K. The hysteresis loops were obtained with a maximum field of 7 × 10^4^ Oe. The samples were previously lyophilized.

**Figure 5 polymers-13-01704-f005:**
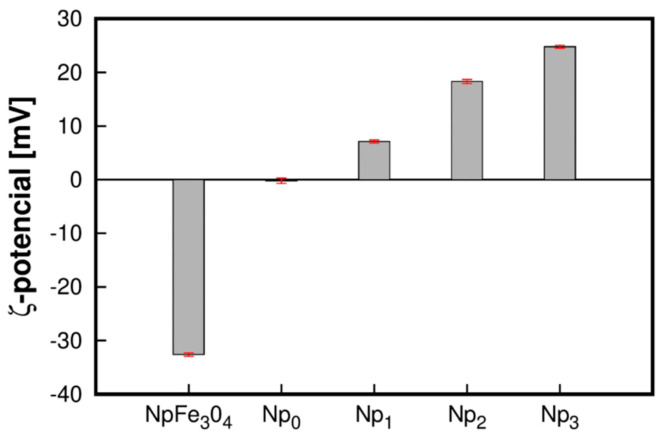
ζ-Potential of Fe_3_O_4_ Nps and samples of Np_0_, Np_1_, Np_2_, and Np_3_, determined by sizer-nano ZS.

**Figure 6 polymers-13-01704-f006:**
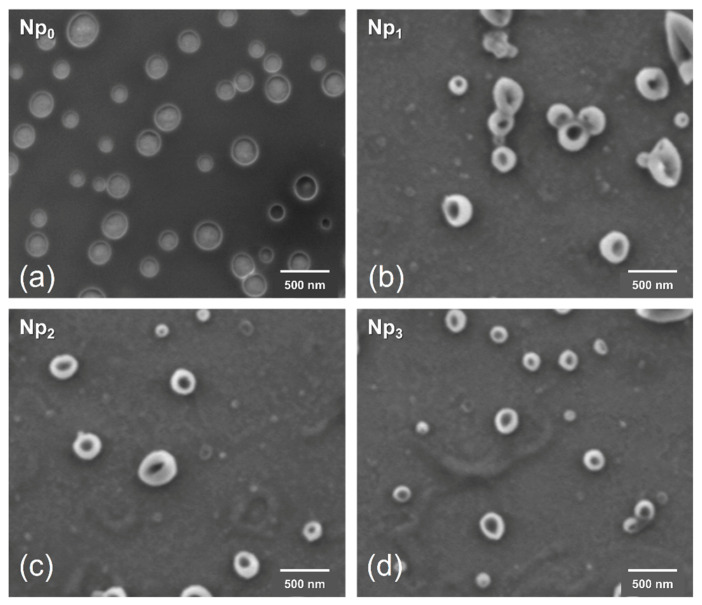
The behavior of size averages as a function of CS concentration. SEM imagen for (**a**) Np_0_; (**b**) Np_1_; (**c**) Np_2_ and (**d**) Np_3_ samples (scale bar = 500 nm).

**Figure 7 polymers-13-01704-f007:**
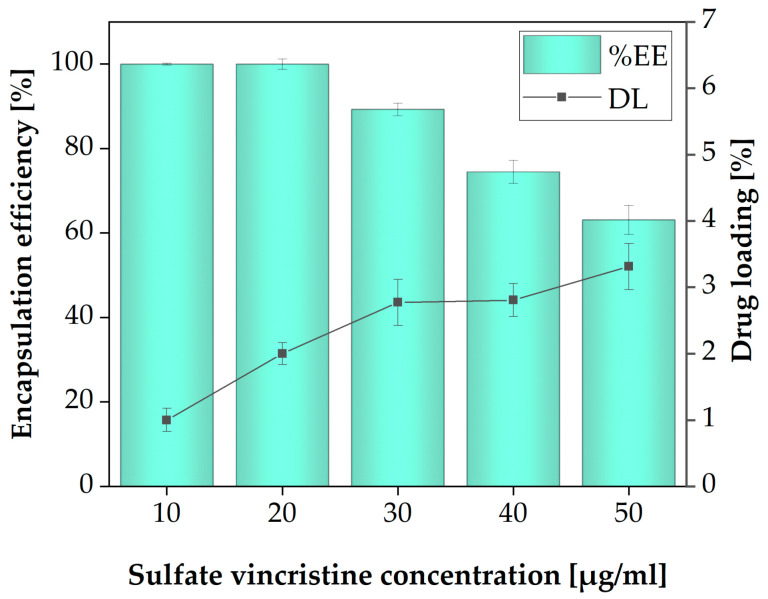
Sulfate vincristine loading and encapsulation efficiency of sample Np_3_.

**Figure 8 polymers-13-01704-f008:**
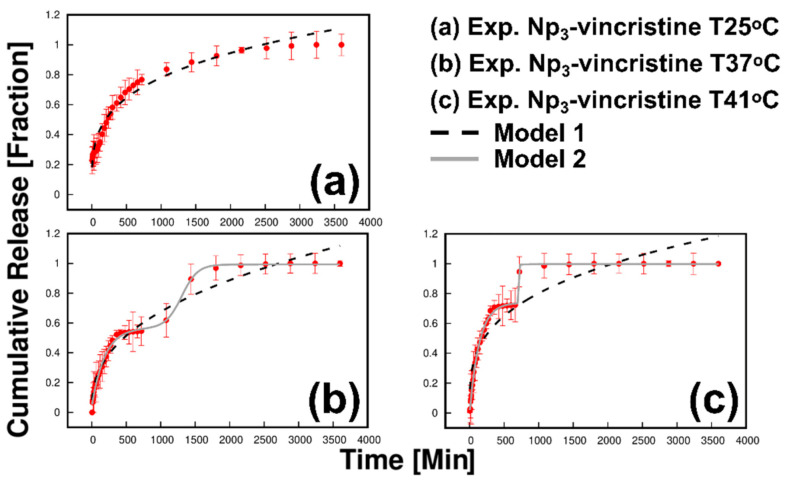
Correlation of mathematical models 1 and 2 of the release fraction of vincristine of the Np_3_-vincristine sample at different temperatures (**a**) 25 °C; (**b**) 37 °C and (**c**) 41 °C.

**Table 1 polymers-13-01704-t001:** Parameters of model 1 for the power-law equation, a classical diffusion Fickian-type.

Temperature (°C)	*K* _1_	*N*	R^2^
25	0.1168	0.2749	0.9754
37	0.0447	0.3929	0.9781
41	0.0985	0.3035	0.8911

**Table 2 polymers-13-01704-t002:** Parameters of model 2, the two-stage model (BiDoseResp).

Temperature (°C)	*A* _1_	*A* _2_	*Logx*01	*Logx*02	*h* _1_	*h* _2_	R^2^
25	-	-	-	-	-	-	-
37	−9.7686	0.9931	−526.4825	1304.0414	0.0037	0.0037	0.99639
41	−66.5933	0.9977	−619.7001	715.7139	0.1503	0.1503	0.99853

## Data Availability

The data presented in this study are available on request from the corresponding author.

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
