# Peer review of "Synthesis and Characterization of a Fe3O4@PNIPAM-Chitosan Nanocomposite and Its Potential Application in Vincristine Delivery"

_polymers, 2021, doi:10.3390/polym13111704_

Round 1

Reviewer 1 Report

In this manuscript, the authors described the synthesis of Fe3O4@PNIPAM-CS and performed the application of vincristine sulfate loading and release. This work is established and organized well, and the results are interesting. so this reviewer recommends the manuscript can be accepted after minor revision.

The authors are advised to consider these points.

  1. In Page 5, line 196, the form of “-NH2-NH3+” seems like incorrect.
  2. The authors should add error bars in Figure 3.
  3. In Figure 4, the authors are encouraged to describe the magnetic curve of NpFe3O4 and Np3 in one graphic.
  4. The authors confirmed the sample Np3 has a well-defined polymeric shell, some TEM characterizations should be conducted to confirm the result. In addition, IR analysis of the samples should be provided.
  5. Abbreviations appeared the first time should be described in full name, such as “NPs” and “AO”.

Author Response

We welcome your comments and feedback.

The modifications made are shown in the attached file.

Reviewer 2 Report

In this study, the authors prepared a core-shell thermomagnetic platform for drug delivery. In general, the article can be reconsidered for publication after a major revision only. 

1) The formation of core-shell structure should be confirmed by TEM study. 

2) FTIR analysis should be performed for all samples including bare Fe3O4

3) Dug release properties should be tested at different pH (biologically relevant). 

4) SEM images of samples before and after drug release can be supplied for comparison. 

5) What about the cytotoxicity of the samples? Suggest how these relatively big particles can be extracted from the body? 

6) Some highly relevant studies should be added to the Introduction part to improve the discussion, i.e. 

- Fabrication of bifunctional core-shell Fe3O4 particles coated with ultrathin phosphor layer, Nanoscale Res. Lett. 2013, 8, 357

- Doxorubicin–Gelatin/Fe3O4–Alginate Dual-Layer Magnetic Nanoparticles as Targeted Anticancer Drug Delivery Vehicles, Polymers 2020, 12, 1747

Author Response

(The authors gave the same response as above.)
